# Self-stigma among clients of outpatient psychiatric clinics: A cross-sectional survey

**Ninni Ihalainen**[1,2], **Eliisa Löyttyniemi**[3], **Maritta Välimäki**[1,4,5]*

**1** Department of Nursing Science, University of Turku, Turku, Finland, **2** Turku University Hospital, Turku, Finland, **3** Department of Biostatistics, University of Turku, Turku, Finland, **4** Xiangya Nursing School, Central South University, Changsha, Hunan, China, **5** Xiangya Center for Evidence-Based Practice & Healthcare Innovation: A Joanna Briggs Institute Affiliated Group, Changsha, China

* maritta.vaelimaeki@csu.edu.cn

**Data Availability Statement:** The datasets underlying this study are not publicly available due to the restriction set by the Finnish Data Protection Act (Finlex 1050/2018). In addition, it was stated in the ethical application and approved by The Ethics

## Abstract

Self-stigma is common among people with mental disorders. A large body of research has examined associations between self-stigma and sociodemographic, clinical and psychosocial factors but the results are still conflicting. The aim of this study was to describe self-stigma among persons with affective and psychotic disorders and identify sociodemographic and clinical factors associated with experiences of self-stigma. A cross-sectional survey was performed with Finnish clients (N = 898) at 16 psychiatric clinics using self-reported questionnaires. The data were analyzed using descriptive statistics and with one-way and multi-way analysis of variance (ANOVA). The results showed that clients in community settings experience self-stigma (a total mean SSMIS-SF score of 74.8 [SD 22.3]). Having a diagnosis of an affective disorder, having a long history of mental disorder (>16 years) and the severity of depressive symptoms were the key factors associated with experiences of self-stigma. Clients living in community settings should be assessed regularly for depressive symptoms of mental disorders, and interventions should be conducted, especially at an early stage of the illness, to reduce self-stigma. Factors associated with self-stigma should be taken into account in the future development of interventions to reduce stigma.

## Introduction

Stigma related to mental disorders is a common problem across cultures and societies [1]. Globally, the prevalence of self-stigma ranges from 22–97% for persons with mental disorders [2]. The incidence of self-stigma for persons with severe mental illness has been reported to vary between 27% and 49% [3]. It has been estimated that, in Europe, a fifth of people with bipolar disorder or depression suffer from self-stigma [4]. Self-stigma, also called internalized stigma, is defined as personal internalization of the prejudice leading to self-discrimination [5]. It is also seen as a process in which a person internalizes the prejudice attitudes leading to self-discrimination, recognizes being part of a stigmatized group and becomes aware of the public stigma, agrees with stigma, applies the stigma to themselves, and then experiences harm

Committee of the University of Turku [ID 55/2018] that participants of the study will informed that research material will be treated confidentially. The data will be collected, analyzed and reported anonymously. The data will not be made available to third parties or organizations. The data will only be used for the purpose for which it was collected. Contact information: The Ethics Committee of the University of Turku. Email: eettinen@utu.fi.

**Funding:** This article was partially supported by personal grants (NI) from the University of Turku, Turku University Foundation and the Department of Nursing Science (DPNurs); and the Finnish government research funding (ERVA) by Turku University Hospital and the University of Turku. The funders had no role in study design, data collection and analysis, decision to publish, or preparation of the manuscript.

**Competing interests:** The first author, Ninni Ihalainen, is affiliated with the hospital involved in the study. This does not alter our adherence to PLOS ONE policies on sharing data and materials.

to their self-esteem because of the stigma [6]. As a subjective process, self-stigma can result in negative self-feelings, maladaptive behaviors, or stereotype endorsement [5]. Self-stigma may also negatively impact treatment commitment [7] and health outcomes [8], and further lead to isolation and ostracism [9]. In its worst case, self-stigma can lead to suicidal behavior [10]. Therefore, there is an urgent need to better understand self-stigma in persons with mental health problems.

Well-documented literature has shown that a variety of factors are associated with self-stigma among persons with mental health problems. We systematically sought published literature in PubMed and found two systematic reviews related to the topic. Livingston and Boyd [11] combined 127 articles to assess a statistical relationship between internalized stigma and at least one other variable for adults living with mental illness and conducted a meta-analysis with 45 studies. The review showed that sociodemographic factors were neither consistently nor strongly correlated with levels of internalized stigma. However, the relationships between internalized stigma and a range of clinical factors such as psychiatric symptom severity, diagnosis, hospitalizations, illness duration, insight, treatment adherence, treatment setting, functioning, medication side effects, were mixed. Symptom severity had a statistically significant and positive association with internalized stigma in 83.3% of the 50 studies. Gerlinger et al. [3] found 5,871 studies related to stigma: 54 studies fulfilled the inclusion criteria, and 44 studies reported on correlates of perceived or experienced stigmatization or self-stigma. Again, the authors reported that socio-demographic factors were only marginally associated with stigma, while psychosocial factors, such as a lower quality of life, showed significant correlations. Illness-related factors showed heterogeneous association with self-stigma. In the review by Gerlinger et al. [3], the prevalence and impact of personal stigma on individual outcomes among schizophrenia spectrum disorder patients were well characterized.

We found 15 recent empirical studies regarding sociodemographic factors associated to self-stigma, the results of which are controversial and inconclusive. For example, some studies reported a positive association between self-stigma and younger age [12, 13] while other studies reported a missing association here [2, 14–21]. The female gender was associated with self-stigma [12, 22], but these results were not confirmed in other studies [2, 4, 13–21, 23]. Further, unemployment [4, 12, 14–16, 18, 20] and lower education levels [4, 15, 17] have been associated with self-stigma, although opposite results have also been reported [12]. In addition, living in a sheltered house [21] or with a nuclear family [12] has found to be associated with higher self-stigma scores, although mixed results can be found [2, 13, 22]. On the contrary, a consistent pattern was found in the lack of an association between self-stigma and marital status [2, 13, 15, 16, 18, 19, 21, 23].

Further, disease or illness-related factors have also been associated with self-stigma. Patients diagnosed with schizophrenia have had higher self-stigma compared to those with affective, bipolar, neurotic or anxiety disorder [2, 12, 17, 19, 24] or bipolar or borderline personality disorder (vs. anxiety disorders) [14, 24]. Severity of depressive symptoms has also been associated with self-stigma [15, 20, 21], although these findings have not been confirmed in other studies [12, 13]. On the contrary, Dubreuck et al. [22] found that self-stigma was higher among patients with bipolar, major depressive or anxiety disorder compared to patients diagnosed with schizophrenia while in studies by Picco et al. [23] and Kalisova et al. [18], the diagnosis of schizophrenia or anxiety was not associated with self-stigma at all. Further, self-stigma was found to be linked to a younger age of onset [12–14, 17], the duration of mental illness [12, 17, 18, 13, 22], an older age at the time of seeking treatment [22] and a higher number of previous hospitalizations [2, 14, 16, 22, 24].

Despite the many existing studies, knowledge about factors related to self-stigma are still uncertain and partially contradictory. First, most studies in this area target persons with

schizophrenia, and less is known about self-stigma in relation to other mental disorders [2, 3, 19, 20, 22]. Second, in most previous studies, the sample size of the participants has been small, typically between 80 and 280. To better generalize the study results, characteristics of self-stigmatized persons with mental disorders should be further explored with larger sample sizes [3, 12, 19, 20, 24]. Third, the interest of self-stigma has usually focused on persons with depression [15]. However, knowledge about the association between depressive symptoms and self-stigma in other mental disorders is unrepresentative [3, 11, 20]. Further, there is a lack of studies about self-stigma among persons with mental health problems in Finland, in both hospital and community settings [25]. The findings of this study would be important for policy makers to consider; after a broad mental health reform in Finland in the 1990s, most persons with mental health problems have been, and continue to be, treated in outpatient clinics [25].

To better understand self-stigma among persons living in a community, we administered a survey related to self-stigma among persons with affective disorders and psychotic disorders to determine the association of self-stigma with sociodemographic and clinical factors. The study results can form a groundwork for further large-scale studies related to stigma in persons with mental health problems in Finland.

## Materials and methods

### Design

A cross-sectional descriptive design with a self-administered survey instrument was used. The design was suitable for our purposes as we examined the phenomenon of self-stigma and relationships among self-stigma at a fixed point in time [26].

### Setting

The study was conducted at 16 outpatient clinics that are part of one hospital district in southern Finland. The hospital district currently serves 479,341 inhabitants in its area [27]. The specific hospital district was selected as it represents a typical Finnish hospital district including specialized psychiatric care. The outpatient clinics were selected as they offer specialized psychiatric treatment focused on affective disorders or psychotic disorders. In general, primary health care services are responsible for mental health care. In more serious cases, clients are treated in the outpatient clinics or psychiatric wards for specialized health care with a referral by a medical doctor at a primary health care center or in private health care services [28]. In outpatient clinics, clients have monthly appointments with their contact person, typically with a psychiatric nurse. Appointments include interactive discussions with therapeutic and supportive approaches or group activities such as with functional and peer groups.

### Target population and eligibility criteria

Our target population included adult clients with mental disorders who visited outpatient clinics specialized in affective disorders and psychotic disorders. This diagnosis group was selected as, based on Finnish health statistics, both affective and psychotic disorders are the major groups of mental disorders in specialized psychiatric outpatient services in Finland [27]. In 2020, there were altogether 200,112 clients in specialized psychiatric outpatient care services: 69,761 (35%) clients were diagnosed with affective disorder and 22,940 (11%) with psychotic disorder [29].

The inclusion criteria were as follows: 1) the person had contact with an outpatient clinic offering services for clients with affective disorders and psychotic disorders 2) the person was between 18 and 65 years old, and 3) the person had an appointment at the outpatient clinic

during the data collection period. Persons were excluded if they were admitted to the hospital during the data collection period or if they were a minor or elderly.

## Sampling method and patient recruitment

A consecutive sampling method was used to ensure the representativeness of our sample [30]. This method is suitable for our purposes as all clients available at a specific location and specific time were invited to participate in this study [31]. Client recruitment was conducted in three phases. First, the researcher contacted the Head Nurses responsible for client services and arranged meetings for the staff working at the clinics. Second, the client recruitment process was introduced to the staff members during the meetings, and staff who were not present at the meeting were informed about the study by the head nurse and the nurses involved. Third, when clients visited appointments in the clinic, nurses assessed if they fulfilled the inclusion criteria. If they did, a nurse offered the client oral and written information about the study during an appointment visit. No formal capacity test was implemented to assess patients' insight; rather, we relied on the judgment of the health care professionals who met with the patients regularly. If clients were willing to participate in the study, they received an unanswered survey form and envelope. Voluntary participation in an anonymous survey was considered as consent to join the study according to Finnish policy and therefore written informed consent was not sought [32–34].

## Instruments

Sociodemographic characteristics: age (years), gender (male, female, other), marital status (single, cohabiting/married, divorced, widowed), education (basic, general/vocational, higher), employment (employed/student, unemployed, rehabilitation support/sick leave, retired, other) and living situation (living alone, living with family, living with relatives or friends, living supported or at half-way home).

Clinical characteristics: a type of mental disorder (affective disorder, psychotic disorder or other, if a respondent felt that their mental disorder could not be categorized as affective or psychotic). In addition, the length of mental disorder (years), the length of psychiatric hospital care (years), and the length of outpatient care (years) was asked.

The Self-Stigma of Mental Illness Scale–Short Form (SSMIS-SF [6]; originally The Self-Stigma of Mental Illness Scale SSMIS [5] is a self-reported questionnaire measuring the level of self-stigma of people with mental illness. The instrument consists of 20 items that form four subscales: 1) Awareness: people are aware of the stereotypes about mental illnesses; 2) Agreement: they agree with these stereotypes; 3) Application: they apply the stereotypes to themselves and, as a result 4) Harm to self-esteem: they experience harm in the form of loss of self-esteem due to their concurrence with the stereotypes [6]. Respondents were asked to respond to the survey using a 9-point agreement scale (1 = I strongly disagree; 9 = I strongly agree). The score of each subscale ranged from 5 to 45, while the total score ranged between 20 and 180; the higher the score, the more self-stigma is endorsed [6]. The instrument had already been translated into Swedish [35], Czech [18], and the Cronbach's alpha value has ranged from 0.62 to 0.90 [6, 34–37]. In this study, the Cronbach's alpha value for each scale ranged from 0.68 to 0.79, and for the total scale value, it was 0.85.

In this study, the SSMIS-SF instrument was used for the first time in Finnish. First, after we received permission to use the instrument, it was translated from English into Finnish using the standard translation-back translation method [38]. Second, the Finnish version was back translated into English by a professional translator not involved in the first phase of the translation. Third, the developer of the instrument checked if the back-

translated version of the instrument corresponded to the original instrument. The Finnish instrument was piloted in two psychiatric wards (N = 18) with volunteer patients (not participating in the main study). Respondents also evaluated the clarity and comprehensibility of the items, instructions for completion and response time [31]. Based on the evaluations, the instrument was clear and easy to complete in 5–10 minutes, and therefore, no further modifications were made.

Depressive symptoms: The Patient Health Questionnaire (PHQ-9 [39]) [40] is a self-reported questionnaire originally developed in English and for use in primary care for depression risk groups to measure a patient's mood [39]. For this study, the Finnish language version of the PHQ-9 instrument, available online, was used. The instrument includes nine items indicating, on a four-point scale, how often they have been concerned by any of the mentioned problems (e.g., fatigue, self-harm) in the past two weeks (0 = not at all; 3 = nearly every day). The higher the score, the more severe the depression symptoms are (range 0–27). Scores are divided into five categories: none (0–4), mild (5–9), moderate (10–14), moderately severe (15–19) and severe (20–27). The PHQ-9 is tested for validity and reliability in a range of languages, including for example, Thai [41] and Greek [42]. The questionnaire has been evaluated to be reliable; PHQ scores $\geq$ 10 have a sensitivity of 88% and a specificity of 88% for major depression [43]. The Cronbach's alpha value has ranged from 0.70 to 0.87 [41, 42]. In this study, the Cronbach's alpha value for each variable ranged from 0.85 to 0.87; for the total scale value, it was 0.88.

## Data collection

The ethical assessment for the study was done by The Ethics Committee of the University of Turku (ID 55/2018, 17 December 2018). The permission to conduct the study was granted by the director of psychiatric specialized health care (ID T09/002/19, 5 March 2019).

The data were collected between 1 April and 10 May 2019 in three different waves, with a one-month response time at each clinic. The respondents filled out the paper questionnaires during their regular appointment with a nurse. We chose to use paper questionnaires to avoid complicating training requirements for the data collection [44]. Completed forms were sealed in an envelope and returned to a sealed box at the clinic. The researcher retrieved the filled envelopes after the data collection period. Two emails were sent to head nurses to remind nurses of the data collection.

A total of 1,364 clients were invited to participate in the study during their outpatient visit; 926 returned a closed envelope with a completed survey form. As eight clients did not meet the eligibility criteria due to the age limit (respondents were over 65 years old) and 20 returned uncompleted questionnaires, we were left with 898 questionnaires to be analyzed (response rate 66%).

## Statistical methods

Categorical variables were first summarized with counts and percentages, continuous variables with means and with standard deviation (SD) and added to the median and range (minimum and maximum). Second, associations between self-stigma (total and subscales) and age (categorized), duration of disorder (categorized), living situation, education, employment status, mental disorder, duration of psychiatric hospital care (categorized), duration of outpatient care (categorized), and depressive symptoms were analyzed first with one-way analysis of variance (ANOVA) (univariate approach). If a factor was statistically significant, the results of the pairwise comparisons were corrected using Tukey's method. Assumptions were checked using studentized residuals. Modelling was continued with multi-way ANOVA including all

significant factors from univariate approach. Then non-significant terms were removed from the model one by one. We did not allow any missing values when calculating the total or sub-scales. Third, the Cronbach's alpha was calculated for all self-stigma subscales. All statistical tests were performed as 2-sided, with a significance level set at 0.05. The data-entering process was done with SPSS (version 25), and the analyses were performed using SAS software, version 9.4 for Windows (SAS Institute Inc., Cary, NC, USA).

# Results

## Background characteristics

The mean age of the respondents (N = 898) was 38.4 years (SD 12.4), and two-thirds were female (N = 562/894, 63%). Detailed characteristics of the respondents are described in Table 1.

## Self-stigma among people with mental disorders

The respondents reported their awareness of stereotypes about mental illnesses. The mean value was highest among awareness in stigma (mean 27.3 [SD 8.3]). The respondents reported how they agreed with these stigmatizing stereotypes of mental illnesses (mean 18.6 [SD 7.5]) and how they applied the stereotypes to themselves (mean 14.3 [SD 7.0]). The final stage of self-stigma suggests that respondents experience harm in the form of a loss of self-esteem due to stereotypes (mean 14.6 [SD 8.2]). The total mean SSMIS-SF score of the respondents was 74.8 (SD 22.3) (Table 2).

**Table 1. Sociodemographic and clinical characteristics of all the respondents (N = 898).**

|  | N | % |
|---|---|---|
| **Age (years) (N = 897)** | | |
| **Min, Mean (SD), Med, Max** | | |
| 18, 38.4 (12.4), 37, 65 | | |
| <25 | 150 | 17 |
| 25–34 | 240 | 27 |
| 35–44 | 210 | 23 |
| 45–54 | 169 | 19 |
| 55< | 128 | 14 |
| **Gender (N = 894)** | | |
| Male | 327 | 36 |
| Female | 562 | 63 |
| Other | 5 | 1 |
| **Marital status (N = 898)** | | |
| Single | 414 | 46 |
| Cohabiting/Married | 344 | 38 |
| Divorced | 127 | 14 |
| Widowed | 13 | 2 |
| **Education (N = 893)** | | |
| Basic | 141 | 16 |
| General/Vocational | 501 | 56 |
| Higher | 251 | 28 |

*(Continued)*

**Table 1.** (Continued)

|  | N | % |
|---|---|---|
| **Employment status (N = 897)** | | |
| Employed/Student | 313 | 35 |
| Unemployed | 154 | 17 |
| Rehabilitation support/Sick leave | 115 | 13 |
| Retired | 264 | 29 |
| Other | 51 | 6 |
| **Living situation (N = 898)** | | |
| Living alone | 422 | 47 |
| Living with family | 439 | 49 |
| Living with relatives or friends | 19 | 2 |
| Living supported or at half-way home | 18 | 2 |
| **Mental disorder (primary) (N = 895)** | | |
| Affective disorder | 636 | 71 |
| Psychotic disorder | 242 | 27 |
| Other | 17 | 2 |
| **Mental disorder (length) (N = 866)** | | |
| **Min, Mean (SD), Med, Max** | | |
| 0, 140.8 (114.7), 120, 720 (months) | | |
| Under 5 years | 221 | 26 |
| 5–15 years | 408 | 47 |
| 16–25 years | 177 | 20 |
| Over 25 years | 60 | 7 |
| **Psychiatric hospital care (N = 889)** | | |
| **Min, Mean (SD), Med, Max** | | |
| 0, 21.8 (51.0), 6, 444 (months) | | |
| Yes | 447 | 50 |
| Length (N = 388) | | |
| Under 1 year | 229 | 59 |
| 1–5 years | 129 | 33 |
| 6–10 years | 18 | 5 |
| Over 10 years | 12 | 3 |
| No | 442 | 50 |
| **Outpatient care (length) (N = 846)** | | |
| **Min, Mean (SD), Med, Max** | | |
| 0, 83.8 (84.8), 48, 540 (months) | | |
| Under 1 year | 99 | 12 |
| 1–5 years | 374 | 44 |
| 6–10 years | 187 | 22 |
| Over 10 years | 186 | 22 |
| **PHQ-9 (N = 888)** | | |
| **Min, Mean (SD), Med, Max** | | |
| 0, 11.3 (6.7), 11, 27 | | |
| None (0–4) | 143 | 16 |
| Mild (5–9) | 256 | 29 |
| Moderate (10–14) | 204 | 23 |
| Moderately severe (15–19) | 164 | 18 |
| Severe (20–27) | 121 | 14 |

**Table 2. Mean (SD), minimum, median scores, maximum and Cronbach's alpha for each subscale in self-stigma for people with mental disorders using the SSMIS-SF.**

| Subscale* | N | Mean (SD) | Min** | Med | Max** | α |
|---|---|---|---|---|---|---|
| Awareness | 892 | 27.3 (8.3) | 5 | 28 | 45 | 0.79 |
| Agreement | 888 | 18.6 (7.5) | 5 | 18 | 45 | 0.74 |
| Application | 897 | 14.3 (7.0) | 5 | 13 | 38 | 0.68 |
| Harm to self-esteem | 894 | 14.6 (8.2) | 5 | 13 | 45 | 0.76 |
| Total | 877 | 74.8 (22.3) | 20 | 75 | 153 | 0.85 |

* The higher the score, the more self-stigma is endorsed for each subscale.

** Subscales for Min can be min 5 and max 45, and subscales for Max can be min 45 and max 180.

## Associations between self-stigma and background characteristics among people with mental illness

**Awareness.** Two statistically significant associations were found regarding awareness of stereotypes. First, those respondents who lived with their family, reported a higher score for awareness of stereotypes compared with respondents who lived alone (mean 28.1 [SD 8.2] vs 26.4 [SD 8.2], $p = 0.017$). Second, those respondents who had an affective disorder reported higher score for awareness than patients with psychotic disorder or other mental disorder (mean 28.2 [SD 8.1] vs 25.4 [SD 8.5] and 22.2 [8.0], all $p < 0.05$).

**Agreement.** Only one statistically significant association was found regarding agreement. Those respondents who had suffered from a mental disorder for 16–25 years or over 25 years reported a higher score for agreement of stigmatized stereotypes than those who had lived with a mental disorder less than five years (mean 19.6 [SD 7.9] and 20.6 [SD 8.5] vs 18.1 [SD 7.2], all $p < 0.05$).

**Application.** Several statistically associations were found regarding the application of stereotypes to themselves. First, a lower education level was linked with a higher application score (all $p < 0.05$). Second, persons who were unemployed, were receiving rehabilitation support/sick leave or retired reported higher scores for application compared to participants who were employed or students (mean 15.2 [SD 6.9]) and 15.5 [SD 6.3] and 14.8 [SD 7.2] vs 13.1 [SD 6.9], all $p < 0.05$). Third, respondents who lived with support or at a half-way home reported higher scores for application compared to respondents who lived alone, with family or with relatives or friends (mean 18.3 [SD 7.8] vs 14.7 [SD 7.9] and 13.9 [SD 7.2] and 12.6 [SD 5.4], all $p < 0.05$). Fourth, as the length of the mental disorder increased, the application score was higher (all $p < 0.05$). Last, respondents who had been in outpatient care for more than 10 years reported higher scores for application than lengths under 1 year or 1–5 years (mean 15.3 [SD 7.2] vs 13.3 [SD 7.0] and 13.8 [SD 7.1], all $p < 0.05$).

**Harm to self-esteem.** Several statistically significant associations were found regarding harm to self-esteem due to stereotypes. First, those who were unemployed or were receiving rehabilitation support/sick leave reported a higher score for harm to self-esteem than those who were employed or students (mean 16.4 [SD 8.6] and 16.9 [SD 7.9] vs 13.2 [SD 7.8], all $p < 0.001$). In addition, those who were unemployed and those receiving rehabilitation/sick leave reported higher score for harm to self-esteem than retired respondents (mean 16.4 [SD 8.6] and 16.9 [SD7.9] vs 14.4 [SD 8.1], all $p < 0.05$). Second, respondents who lived with support or at a half-way home reported higher scores for harm to self-esteem than respondents who lived alone, with family or with relatives or friends (mean 20.4 [SD 9.7] vs 14.6 [SD 7.9] and 14.4 [SD 8.4] and 15.1 [SD 6.5], all $p < 0.05$). Third, those respondents who had an

affective disorder reported higher scores for harm to self-esteem than respondents with psychotic disorder (mean 15.3 [SD 8.3] vs 13.3 [SD 7.7], p < 0.05).

**Total score for self-stigma.**   Regarding the total score for self-stigma, we found statistically significant associations in the results depending on the type of mental disorder present and its length. First, respondents who identified as having an affective disorder reported higher total scores of self-stigma than respondents with psychotic disorder (mean 76.9 [SD 22.2] vs 70.0 [SD 21.7], p < 0.001). Second, respondents who had suffered from a mental disorder 16–25 years or over 25 years reported higher total scores for self-stigma than those who had lived with mental illness for less than five years (mean 78.0 [SD 23.1] and 79.0 [SD 23.4] vs 72.3 [SD 21.8], all p < 0.05) (Table 3).

Other background factors, such as age, gender, marital status, and psychiatric hospital care, were not significantly associated with any SSMIS-SF subscales or total scores (all p-values > 0.05).

### Association with depressive symptoms and self-stigma

Depressive symptoms, handled as a categorized PHQ-9 scale, were significant in all categories in the total level of self-stigma and in all subscales (p < 0.001). If respondents had higher mean values with depressive symptoms, they experienced a higher level of self-stigma (Table 3).

Further, when the final model in multivariable approach was conducted, we found that length of mental disorder (p = 0.018) and PHQ-9 (categorized) (p < 0.001) were significantly associated to self-stigma. This means that the longer the respondent had had the mental disorder and the more severe their depressive symptoms were, the higher the total score of self-stigma was (Table 4). Similarly, multivariable models were programmed to self-stigma subscales, and the results were quite in line with univariate approach. The greatest exception was that type of mental disorder dropped from two final models due strong association with PHQ-9 (Table 5).

### Discussion

This study aimed to describe self-stigma among persons with affective and psychotic disorders in outpatient psychiatric care. We also identified associations between self-stigma and sociodemographic and clinical factors. To the best of our knowledge, this is the first study exploring the relationship between self-stigma and these associated factors in Finnish outpatient services. This topic is indeed important in a high-income welfare country like in Finland as our results show that persons with affective and psychotic disorders do internalize public stereotypes, which can lead to experiences of self-stigma.

Gerlinger et al. [3] in their systematic review found that socio-demographic factors were only marginally associated with personal stigma. Our study results also support these findings as most sociodemographic factors did not show a statistically significant association with self-stigma. On the other hand, Gerlinger et al. [3] concluded that there were few significant or contradictory associations between age of onset, duration of illness and personal stigma, and the authors proposed the need to examine the topic further. Therefore, we did continue to examine this topic and found that the diagnosis of affective disorder was associated with higher scores of self-stigma than the psychotic disorder was. As far as we are aware, a limited number of studies have reported that persons with affective disorders (bipolar, depressive and anxiety disorder) have a higher sense of self-stigma than persons with schizophrenia [22]. In general, most studies related to self-stigma have been conducted among patients with schizophrenia [2, 3, 12, 17, 19, 24], and even fewer studies have compared self-stigma among different diagnosis groups [2, 3, 19, 20, 22].

**Table 3. Mean (SD) for each subscale in self-stigma for people with mental disorders (N = 898) using the SSMIS-SF in characteristics and depressive symptoms associated with self-stigma.**

| | N | Awareness | Agreement | Application | Harm to self-esteem | Total |
|---|---|---|---|---|---|---|
| | | | | | | Mean (SD) |
| **Education***  | | | | | | |
| Basic | 141 | 27.3 (8.0) | 19.4 (8.2) | 15.7 (7.7) | 15.3 (9.3) | 77.4 (24.9) |
| General/Vocational | 501 | 26.7 (8.2) | 18.8 (7.4) | 14.4 (7.1) | 14.7 (7.8) | 74.6 (22.2) |
| Higher | 251 | 28.2 (8.5) | 17.7 (7.1) | 13.2 (6.2) | 13.9 (7.8) | 73.2 (20.5) |
| **p-value** | | 0.066 | 0.060 | **0.004** | 0.204 | 0.204 |
| **Employment status***  | | | | | | |
| Employed/student | 313 | 27.9 (8.2) | 18.5 (7.1) | 13.1 (6.9) | 13.2 (7.8) | 72.7 (21.6) |
| Unemployed | 154 | 26.7 (8.3) | 18.5 (7.5) | 15.2 (6.9) | 16.4 (8.6) | 75.9 (22.6) |
| Rehab. support/sick leave | 115 | 28.5 (7.5) | 19.0 (8.0) | 15.5 (6.3) | 16.9 (7.9) | 79.9 (21.0) |
| Retired | 264 | 26.4 (8.7) | 18.7 (7.9) | 14.8 (7.2) | 14.4 (8.1) | 74.6 (23.3) |
| Other | 51 | 27.0 (8.4) | 18.5 (7.0) | 14.3 (7.6) | 14.5 (8.4) | 74.0 (22.2) |
| **p-value** | | 0.104 | 0.961 | **0.002** | **<0.001** | 0.055 |
| **Living situation***  | | | | | | |
| Alone | 422 | 26.4 (8.2) | 18.9 (7.5) | 14.7 (7.9) | 14.6 (7.9) | 74.2 (22.1) |
| With family | 439 | 28.1 (8.2) | 18.5 (7.4) | 13.9 (7.2) | 14.4 (8.4) | 75.1 (22.4) |
| With relatives/friends | 19 | 26.9 (7.7.) | 16.2 (6.7) | 12.6 (5.4) | 15.1 (6.5) | 71.0 (17.0) |
| Supported/at half-way home | 18 | 28.7 (9.7) | 18.4 (10.5) | 18.3 (7.8) | 20.4 (9.7) | 85.7 (26.9) |
| **p-value** | | **0.017** | 0.567 | **0.045** | **0.036** | 0.281 |
| **Mental disorder***  | | | | | | |
| Affective disorder | 636 | 28.2 (8.1) | 18.8 (7.6) | 14.6 (7.1) | 15.3 (8.3) | 76.9 (22.2) |
| Psychotic disorder | 242 | 25.4 (8.5) | 18.0 (7.3) | 13.8 (6.9) | 13.3 (7.7) | 70.0 (21.7) |
| Other | 17 | 22.2 (8.0) | 20.7 (7.1) | 13.4 (4.9) | 12.6 (6.3) | 68.2 (20.8) |
| **p-value** | | **<0.001** | 0.178 | 0.258 | **0.003** | **<0.001** |
| **Mental disorder (length)***  | | | | | | |
| Under 5 years | 218 | 27.1 (8.3) | 18.1 (7.2) | 13.1 (7.0) | 13.9 (7.7) | 72.3 (21.8) |
| 5–15 years | 400 | 26.9 (8.2) | 18.1 (7.3) | 14.3 (6.5) | 14.7 (7.9) | 74.1 (21.6) |
| 16–25 years | 176 | 28.0 (8.0) | 19.6 (7.9) | 15.3 (7.5) | 15.2 (9.3) | 78.0 (23.1) |
| Over 25 years | 56 | 28.4 (9.5) | 20.6 (8.5) | 15.4 (7.6) | 14.6 (7.7) | 79.0 (23.4) |
| **p-value** | | 0.372 | **0.020** | **0.007** | 0.481 | **0.031** |
| **Outpatient care (length)***  | | | | | | |
| Under 1 year | 99 | 28.6 (8.3) | 18.8 (7.2) | 13.3 (7.0) | 14.0 (8.1) | 74.6 (21.7) |
| 1–5 years | 374 | 26.7 (8.4) | 18.1 (7.2) | 13.8 (7.1) | 14.9 (8.1) | 73.3 (22.5) |
| 6–10 years | 187 | 27.1 (8.0) | 18.5 (7.4) | 14.8 (6.6) | 15.1 (8.4) | 75.6 (21.9) |
| Over 10 years | 186 | 27.9 (8.4) | 19.6 (8.1) | 15.3 (7.2) | 14.2 (8.3) | 77.0 (22.7) |
| **p-value** | | 0.154 | 0.138 | **0.024** | 0.537 | 0.291 |
| **PHQ-9 (categorized)***  | | | | | | |
| None (0–4) | 143 | 24.2 (8.8) | 16.2 (7.1) | 10.1 (5.9) | 9.0 (5.9) | 59.7 (19.7) |
| Mild (5–9) | 256 | 27.3 (8.1) | 18.3 (7.4) | 12.3 (5.9) | 11.5 (6.6) | 69.6 (20.0) |
| Moderate (10–14) | 204 | 26.8 (8.1) | 18.5 (6.8) | 15.3 (6.7) | 16.0 (7.1) | 76.5 (19.6) |
| Moderately severe (15–19) | 164 | 28.2 (7.9) | 20.1 (8.0) | 16.9 (6.9) | 17.7 (8.3) | 82.4 (21.8) |
| Severe (20–27) | 121 | 30.3 (7.8) | 20.2 (7.9) | 18.2 (7.0) | 21.2 (8.0) | 89.8 (20.3) |
| **p-value** | | **<0.001** | **<0.001** | **<0.001** | **<0.001** | **<0.001** |

* The higher the score, the more self-stigma is endorsed for each subscale.

More detailed statistical results (pairwise comparisons) from Table 3 are presented in S1 Table.

**Table 4. Final multivariable model for self-stigma of Mental Illness Scale and all subscales.** All significant explanatory variables from univariate modelling were first included and then non-significant terms were removed.

| | Awareness | Agreement | Application | Harm to self-esteem | Total |
|---|---|---|---|---|---|
| | Model-based means (SE) | | | | |
| **Employment status** | | | | | |
| Employed/student | | | | 14.0 (0.6) | |
| Unemployed | | | | 15.8 (0.6) | |
| Rehab. support/sick leave | | | | 16.1 (0.7) | |
| Retired | | | | 15.3 (0.4) | |
| Other | | | | 15.9 (1.0) | |
| **p-value** | | | | **0.024** | |
| **Mental disorder** | | | | | |
| Affective disorder | 28.0 (0.3) | | | | |
| Psychotic disorder | 26.2 (0.6) | | | | |
| Other | 22.2 (2.0) | | | | |
| **p-value** | **0.007** | | | | |
| **Mental disorder (length)** | | | | | |
| Under 5 years | | 18.2 (0.5) | | | 73.4 (1.4) |
| 5–15 years | | 18.1 (0.4) | | | 74.6 (1.0) |
| 16–25 years | | 19.7 (0.6) | | | 79.2 (1.6) |
| Over 25 years | | 20.5 (1.0) | | | 78.5 (2.7) |
| **p-value** | | **0.019** | | | **0.018** |
| **Outpatient care (length)** | | | | | |
| Under 1 year | | | 13.4 (0.6) | | |
| 1–5 years | | | 13.8 (0.3) | | |
| 6–10 years | | | 15.2 (0.5) | | |
| Over 10 years | | | 16.0 (0.5) | | |
| **p-value** | | | **0.003** | | |
| **PHQ-9 (categorized)** | | | | | |
| None (0–4) | 22.7 (0.9) | 16.7 (0.6) | 10.0 (0.6) | 9.5 (0.6) | 60.8 (1.8) |
| Mild (5–9) | 25.6 (0.8) | 18.9 (0.5) | 12.2 (0.4) | 12.0 (0.5) | 70.7 (1.4) |
| Moderate (10–14) | 24.8 (0.9) | 18.9 (0.6) | 15.3 (0.5) | 16.3 (0.5) | 77.3 (1.6) |
| Moderately severe (15–19) | 26.1 (0.9) | 20.6 (0.6) | 17.3 (0.5) | 18.0 (0.6) | 83.4 (1.7) |
| Severe (20–27) | 28.0 (1.0) | 20.5 (0.7) | 18.2 (0.6) | 21.5 (0.7) | 89.8 (2.0) |
| **p-value** | **<0.001** | **<0.001** | **<0.001** | **<0.001** | **<0.001** |

Table includes model-based means, standard errors (SE) and p-value for the explanatory variable.

**Table 5. Association between PHQ-9 (as categorized) and type of mental disorder.**

| PHQ-9 (categorized) | Affective disorder N (%) | Psychotic disorder N (%) | Other | Total |
|---|---|---|---|---|
| | | | N (%) | N (%) |
| None (0–4) | 71 (11.3) | 69 (28.9) | 2 (11.8) | 142 (16.1) |
| Mild (5–9) | 160 (25.4) | 89 (37.2) | 5 (29.4) | 254 (28.7) |
| Moderate (10–14) | 151 (24.0) | 50 (20.9) | 3 (17.6) | 204 (23.0) |
| Moderately severe (15–19) | 138 (22.0) | 20 (8.4) | 6 (35.3) | 164 (18.5) |
| Severe (20–27) | 109 (17.3) | 11 (4.6) | 1 (5.9) | 121 (13.7) |
| Total | 629 (100) | 239 (100) | 17 (100) | 885 (100) |

We can only speculate here as to why self-stigma is more common among persons with affective disorders than those with psychotic disorders. Our results show that clients with affective disorders were more aware of public stereotypes compared to clients with psychotic disorders. It has been proposed that persons with depression tend to perceive the reactions of their social environment in a negative way, indicating that the perceptions of stigmatization might be a symptom of the underlying pathology rather than an independent variable [3]. We may seek an explanation for our results by looking at different career pathways in society. Both schizophrenia (30%) and depression (29%) are the two most common diagnoses for disability pension in Finland [45]. However, the timespan of careers in these groups is different. For example, the retirement age for persons with depression is 60–64 years [45], and many of these persons have stable working lives in society before retirement. On the contrary, 27% of persons with schizophrenia in Finland retire between the age of 25 and 34 years [45]. Compared to other diagnostic groups, unemployment is the highest among people with schizophrenia, ranging from 89% to 94% [46]. If persons need to give up their work status after a well-established career, it may cause shame [8] and extra pressure due to stigmatized attitudes in the community [20]. Perhaps persons with schizophrenia are more often "used to" public stigma as their illness often appears in early adulthood. Therefore, to prevent self-stigma in any diagnostic group and at any life stage, it is important to be more aware of the factors that may contribute to an increase in the encounter with stigma at work and in society, as well as an increase in self-stigma [47].

We found that the severity of the depressive symptoms in both of our diagnostic groups—affective and psychotic disorder—were positively associated with self-stigma; the finding was supported by previous studies [15, 20, 21]. While severity of the depressive symptoms and PHQ-9 were strongly associated so that in affective disorder the severity was more severe, PHQ-9 seemed to be stronger factor when total self-stigma was modelled. Recently, public awareness of mental disorders, including depression, has increased with many programs and public campaigns [48]. It has also been found that public disclosure of depression and depressive symptoms have increased acceptance attitudes toward depression [49]. Still persons with mental disorders are exposed to negative situations over time, and the risk of experiencing a higher degree of self-stigma also increases [21]. Therefore, the association of depressive symptoms with self-stigma in mental disorder should be further investigated in longitudinal research as studies have shown that both depressive symptoms and self-stigma are associated with suicidal ideation [10, 16, 21]. In addition, our results support previous findings that a long history of mental disorder increases self-stigma, so identification and treatment of self-stigma should be tackled at an early stage of illness [12, 13, 16].

## Limitations

Our study has several limitations. First, the data collection was based on respondents' self-reported information, so, precise information about diagnoses, treatment history and other objective measures could not be collected. We may therefore question the validity of the respondents' own categorization of their diagnostic group. We can also speculate whether participants even had the insight to identify themselves in specific diagnosis category. For example, 17 patients responded with "Other" if they felt that they did not belong to either diagnostic group or if they were unsure to which diagnostic group they belonged. Indeed, it has been shown that some persons with mental illness may not be aware of their own diagnosis or that they lack insight on their illness. We did not use a specific assessment method to assure respondents' mental status or insight for screening purposes during patient recruitment. It would have been helpful to have been able to confirm patients' capacity or insight; not having

been able to caused a potential point of risk. On the other hand, the service system in the study organization is divided into different pathways based on diagnostic groups: one for those with affective disorders and one for those with psychotic disorders. Respondents' categorization of their diagnoses could then be assured by comparing organization records. In addition, we had only 2% of patients who were diagnosed with other diagnoses than affective disorder or psychotic disorder. Therefore, our study results might be generalized in these two specific study groups only.

Second, as the study was cross-sectional, results could only be captured at specific time points. To detect any causal or long-term effects of self-stigma in persons with mental disorders, further studies are needed to answer these questions. Third, this study focused participants' socio-demographic and clinical factors, and lacked an investigation into the association between self-stigma and any psychosocial factors as been recommended by other investigators [11, 18]. To offer a deeper insight into this complex phenomenon, factors related to self-stigma, different career pathways, life stages and psychological factors, such as hope, self-esteem, empowerment and quality of life, should be studied in the future.

Taken all together, despite the limitations, the findings from this study expand current knowledge about self-stigma and its related factors among patients living and treated in a community setting. A more context-oriented research approach with a longitudinal design could bring new understanding to how stigmatized attitudes develop as part of social integration and individual functioning and in specific contexts. Future work should also focus on further exploring the occurrence of self-stigma among vulnerable risk groups that are often excluded in society.

## Conclusions

Self-stigma is prevalent among outpatients living in the community with affective disorders and psychotic disorders in Finland. As having a diagnosis of an affective disorders, long duration of illness and severe depressive symptoms were the key factors in having a sense of self-stigma, special attention is needed on community settings to regularly assess depressive symptoms for mental disorders and to develop and conduct interventions to reduce self-stigma, especially at an early stage of the illness.

## Supporting information

**S1 Table. Overall p-values and also p-values from pairwise comparisons when overall p-value less than 0.05.**
(DOCX)

## Acknowledgments

We appreciate the willing cooperation of the nursing staff during the data collection. We would also like to thank Professor Patrick Corrigan and his assistant for their co-operation in the translation process of the SSMIS instrument. Finally, we thank Leigh Ann Lindholm for language checking.

## Author Contributions

**Conceptualization:** Ninni Ihalainen, Maritta Välimäki.

**Data curation:** Ninni Ihalainen.

**Formal analysis:** Ninni Ihalainen, Eliisa Löyttyniemi.

**Funding acquisition:** Maritta Välimäki.

**Investigation:** Ninni Ihalainen.

**Methodology:** Ninni Ihalainen, Eliisa Löyttyniemi, Maritta Välimäki.

**Project administration:** Ninni Ihalainen, Maritta Välimäki.

**Resources:** Ninni Ihalainen, Maritta Välimäki.

**Software:** Eliisa Löyttyniemi.

**Supervision:** Maritta Välimäki.

**Validation:** Ninni Ihalainen, Eliisa Löyttyniemi, Maritta Välimäki.

**Visualization:** Ninni Ihalainen, Maritta Välimäki.

**Writing – original draft:** Ninni Ihalainen, Eliisa Löyttyniemi, Maritta Välimäki.

**Writing – review & editing:** Ninni Ihalainen, Eliisa Löyttyniemi, Maritta Välimäki.

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
