## [Decision Letter · Decision Letter 0]

1 Dec 2021

PONE-D-21-27914Self-stigma among clients of outpatient psychiatric clinics: A cross-sectional surveyPLOS ONE

Dear Dr. Vällmäki,

Thank you for submitting your manuscript to PLOS ONE. After careful consideration, we feel that it has merit but does not fully meet PLOS ONE’s publication criteria as it currently stands. Therefore, we invite you to submit a revised version of the manuscript that addresses the points raised during the review process.

We look forward to receiving your revised manuscript.

Kind regards,

Stephan Doering, M.D.

Academic Editor

PLOS ONE

“I have read the journal's policy and the authors of this manuscript have the following competing interests: [The first author, Ninni Ihalainen, is afﬁliated with the hospital involved in the study.]”

Reviewers' comments:

Reviewer's Responses to Questions

**Comments to the Author**

1. Is the manuscript technically sound, and do the data support the conclusions?

Reviewer #1: Yes

Reviewer #2: Yes

2. Has the statistical analysis been performed appropriately and rigorously? 

Reviewer #1: I Don't Know

Reviewer #2: Yes

3. Have the authors made all data underlying the findings in their manuscript fully available?

Reviewer #1: No

Reviewer #2: Yes

4. Is the manuscript presented in an intelligible fashion and written in standard English?

Reviewer #1: Yes

Reviewer #2: Yes

5. Review Comments to the Author

Reviewer #1: The article reports findings of cross-sectional survey investigating self-stigma among a convenience sample of outpatients in Finland. The topic is generally interesting, however I identified several issues that should be adressed before publication:

1) In the introduction the authors state that the topic was already investigated in many studies, which had conflicting findings. I agree that there is a need for additional investigations, however in a first step the reasons for the inconsistent findings of past literature should be identified and adressed using more rigorous methods (e.g. longitudinal designs). I am not sure how another cross-sectional study (using a convenience sample) can help to better understand self-stigma among people with mental illness. What new information is provided?

2) The authors should start the introduction with clear definitions of used concepts (i.e. self-stigma, public stigma, personal stigma,...)

3) I suggest to exchange Ref 10 -> the cited article is theoretical and since its publication in 2014 several experimental studies investigated this topic (partly using longitudinal data) which allow to make the statement of an association between self-stigma and suicidality

4) The authors state their data is available within the manuscript and its supporting infromation files. However, the manuscript does not include raw data (it merely reports means etc.) and no other files were available to me.

Reviewer #2: The manuscript is well-written. It gives a detailed picture of self-stigma among psychiatric patients attending outpatient clinics. It highlights that the experiences of self-stigma among mentally ill people are prevalent irrespective of the living place or the country. Also, the exploration of factors associated with self-stigma is similar to previous studies, which suggests that future work should focus on this field to explore further the occurrence of self-stigma among the risk group of people.

Some corrections regarding grammar and typo are suggested in the attached. Also, other suggestions are made which might strengthen the value of the manuscript and add more detail about the experiences of self-stigma among mentally ill patients.

6. PLOS authors have the option to publish the peer review history of their article (what does this mean?). If published, this will include your full peer review and any attached files.

Reviewer #1: No

Reviewer #2: **Yes: **Bimala Panthee

---

## [Author Response · Author response to Decision Letter 0]

27 Jan 2022

Reviewer #1: The article reports findings of cross-sectional survey investigating self-stigma among a convenience sample of outpatients in Finland. The topic is generally interesting, however I identified several issues that should be adressed before publication:

1) In the introduction the authors state that the topic was already investigated in many studies, which had conflicting findings. I agree that there is a need for additional investigations, however in a first step the reasons for the inconsistent findings of past literature should be identified and adressed using more rigorous methods (e.g. longitudinal designs). I am not sure how another cross-sectional study (using a convenience sample) can help to better understand self-stigma among people with mental illness. What new information is provided? 

RESPONSE: We clarified the purpose of the study and added justifications as follows: “Despite the many existing studies, knowledge about factors related to self-stigma are still uncertain and partially contradictory. First, most studies in this area target persons with schizophrenia, and less is known about self-stigma in relation to other mental disorders [2, 3, 19, 20, 22]. Second, in most previous studies, the sample size of the participants has been small, typically between 80 and 280. To better generalize the study results, characteristics of self-stigmatized persons with mental disorders should be further explored with larger sample sizes [3, 12, 19, 20, 24]. Third, the interest of self-stigma has usually focused on persons with depression [15]. However, knowledge about the association between depressive symptoms and self-stigma in other mental disorders is unrepresentative [3, 11, 20]. Further, there is a lack of studies about self-stigma among persons with mental health problems in Finland, in both hospital and community settings [26]. The findings of this study would be important for policy makers to consider; after a broad mental health reform in Finland in the 1990s, most persons with mental health problems have been, and continue to be, treated in outpatient clinics [26]. 

To better understand self-stigma among persons living in a community, we administered a survey related to self-stigma among persons with affective disorders and psychotic disorders to determine the association of self-stigma with sociodemographic and clinical factors. The study results can form a groundwork for further large-scale studies related to stigma in persons with mental health problems in Finland.”

2) The authors should start the introduction with clear definitions of used concepts (i.e. self-stigma, public stigma, personal stigma,...)

RESPONSE: This section has been modified to make it clearer as follows:” Self-stigma, also called internalized stigma, is defined as personal internalization of the prejudice leading to self-discrimination [5]. It is also seen as a process in which a person internalizes the prejudice attitudes leading to self-discrimination, recognizes being part of a stigmatized group and becomes aware of the public stigma, agrees with stigma, applies the stigma to themselves, and then experiences harm to their self-esteem because of the stigma [6].”

3) I suggest to exchange Ref 10 -> the cited article is theoretical and since its publication in 2014 several experimental studies investigated this topic (partly using longitudinal data) which allow to make the statement of an association between self-stigma and suicidality.

RESPONSE: We removed this reference and added a more appropriate one. (Oexle, N., Rüsch, N., Viering, S., Wyss, C., Seifritz, E., Xu, Z., et.al. Self‑stigma and suicidality: a longitudinal study. Eur Arch Psychiatry Clin Neurosci. 2017;267: 359–361.

https://doi.org/10.1007/s00406-016-0698-1)

4) The authors state their data is available within the manuscript and its supporting infromation files. However, the manuscript does not include raw data (it merely reports means etc.) and no other files were available to me. 

RESPONSE: Based on the ethical statement of this study (The Ethics Committee of the University of Turku [ID 55/2018]) and the Finnish Data Protection Act (Finlex 1050/2018) we are not allowed to make data publicly available. 

According to instructions of the journal, we have requested a correction for this in a cover letter. 

In the end of the submitted article more detailed statistical results (pairwise comparisons) from Table 3 are presented in supporting information S1 Table.

Reviewer #2: The manuscript is well-written. It gives a detailed picture of self-stigma among psychiatric patients attending outpatient clinics. It highlights that the experiences of self-stigma among mentally ill people are prevalent irrespective of the living place or the country. Also, the exploration of factors associated with self-stigma is similar to previous studies, which suggests that future work should focus on this field to explore further the occurrence of self-stigma among the risk group of people. Some suggestions mentioned below might strengthen the value of the manuscript and add more detail about the experiences of self-stigma among mentally ill patients.

RESPONSE: The notion of further study needed have been added into the Discussion section. 

Major corrections

1) Rationale of the study: There are plenty of data related to factors associated with self-stigma and its prevalence. However, the knowledge gap and the need for the study in Finland have not been addressed.

RESPONSE: We clarified the knowledge gap and added justifications as follows; “ Despite the many existing studies, knowledge about factors related to self-stigma are still uncertain and partially contradictory. First, most studies in this area target persons with schizophrenia, and less is known about self-stigma in relation to other mental disorders [2, 3, 19, 20, 22]. Second, in most previous studies, the sample size of the participants has been small, typically between 80 and 280. To better generalize the study results, characteristics of self-stigmatized persons with mental disorders should be further explored with larger sample sizes [3, 12, 19, 20, 24]. Third, the interest of self-stigma has usually focused on persons with depression [15]. However, knowledge about the association between depressive symptoms and self-stigma in other mental disorders is unrepresentative [3, 11, 20]. Further, there is a lack of studies about self-stigma among persons with mental health problems in Finland, in both hospital and community settings [26]. The findings of this study would be important for policy makers to consider; after a broad mental health reform in Finland in the 1990s, most persons with mental health problems have been, and continue to be, treated in outpatient clinics [26]. 

To better understand self-stigma among persons living in a community, we administered a survey related to self-stigma among persons with affective disorders and psychotic disorders to determine the association of self-stigma with sociodemographic and clinical factors. The study results can form a groundwork for further large-scale studies related to stigma in persons with mental health problems in Finland.”

2) Line 147: Inclusion criteria: There is no information regarding the insight of the respondents. How did you assume that the respondents had a good insight to answer the questionnaire? Had the insight been checked before distributing the questionnaire? If yes, the detailed information should be included in the “sampling method and patient recruitment” section. If not, it should be mentioned in the study's limitations, which would greatly influence the result of this study.

RESPONSE: We have clarified this information in the “sampling method and patient recruitment” section as follows: “No formal capacity test was implemented to assess patients’ insight; rather, we relied on the judgment of the health care professionals who met with the patients regularly.” 

We have also taken this into account in the section of limitations as follows; “the data collection was based on respondents’ self-reported information, so, precise information about diagnoses, treatment history and other objective measures could not be collected. We may therefore question the validity of the respondents’ own categorization of their diagnostic group. We can also speculate whether participants even had the insight to identify themselves in specific diagnosis category. For example, 17 patients responded with “Other” if they felt that they did not belong to either diagnostic group or if they were unsure to which diagnostic group they belonged. Indeed, it has been shown that some persons with mental illness may not be aware of their own diagnosis or that they lack insight on their illness. We did not use a specific assessment method to assure respondents’ mental status or insight for screening purposes during patient recruitment. It would have been helpful to have been able to confirm patients’ capacity or insight; not having been able to caused a potential point of risk. On the other hand, the service system in the study organization is divided into different pathways based on diagnostic groups: one for those with affective disorders and one for those with psychotic disorders. Respondents’ categorization of their diagnoses could then be assured by comparing organization records. “

3) As found in results, patients with affective disorder had high self-stigma. Did you analyze affective disorder and depressive symptoms?

RESPONSE: We study association between three types of mental disorder and depression as classified into five categories with Fisher's exact test and we found strong association (p<0.0001). In affective disorder group, subjects had clearly more severe depression symptoms than in other two groups. 

4) Population of the study: From the beginning of the abstract, it is clear that the study population is persons with affective and psychotic disorders. However, in the result section, 17 people were categorized as others (diagnosis). Therefore, it is suggested to justify it or remove those 17 and reanalyze. It may affect the final concluding result as explained in the manuscript “affective disorder people have higher experiences of self-stigma”.

RESPONSE: 17 ‘other’ subjects were included in our analyses. We feel that they have minor impact to the results, but it would be important to include all respondents to the analyses. However, if reviewers strongly feels that we should remove those subjects, we are ready to do it.

Minor corrections

5) Line 126: The meaning of statement is not clear. Please rewrite.

RESPONSE: We modified this statement and added new reference as follows:” The design was suitable for our purposes as we examined the phenomenon of self-stigma and relationships among self-stigma at a fixed point in time [27].“

6) Line 130: The statement would be difficult for people outside Finland to understand. Please specify if these 16 outpatient clinics are run by one district hospital or in one district hospital, there are 16 outpatient clinics. Maybe providing brief introduction regarding Finish healthcare would be appropriate.

RESPONSE: This Setting section has been modified to make it more informative as follows; “The study was conducted at 16 outpatient clinics that are part of one hospital district in southern Finland. The hospital district currently serves 479,341 inhabitants in its area [28]. The specific hospital district was selected as it represents a typical Finnish hospital district including specialized psychiatric care. The outpatient clinics were selected as they offer specialized psychiatric treatment focused on affective disorders or psychotic disorders. In general, primary health care services are responsible for mental health care. In more serious cases, clients are treated in the outpatient clinics or psychiatric wards for specialized health care with a referral by a medical doctor at a primary health care center or in private health care services [29]. In outpatient clinics, clients have monthly appointments with their contact person, typically with a psychiatric nurse. Appointments include interactive discussions with therapeutic and supportive approaches or group activities such as with functional and peer groups.“ 

7) Clinical characteristics: type of mental disorder (other) should be clarified as mentioned above 

RESPONSE: We clarified ”Other” in the sections of methods as follows; “Clinical characteristics: a type of mental disorder (affective disorder, psychotic disorder or other, if a respondent felt that their mental disorder could not be categorized as affective or psychotic). 

We clarified this in Limitations also.

8) Instruments: PHQ-9 Please indicate the language of the questionnaire used. 

RESPONSE: We added the sentence as follows: ” For this study, the Finnish language version of the PHQ-9 instrument, available online, was used.”

9) What was the validity and reliability of this tool in Finish sample?

RESPONSE: We added the values of the Cronbach’s alpha for each variable (ranged from 0.85 to 0.87), and for the total scale value (0.88) in the Instruments section.

10) Line 246: Suggested “respondents” despite “participants” throughout the manuscript as it is a quantitative study.

RESPONSE: Throughout the manuscript ”participants” has been removed and ”respondents” has been added instead. 

Table 1 (variables) 

11) Mental disorder (length): 5-15 years Hyphen is missing

RESPONSE: A hyphen has been added to the table.

12) Psychiatric hospital care: It appears that only 388 out of 447 reported the duration. Please move 388 to column 1 for clarity

RESPONSE: The number 388 has been moved to column 1 in table 1. 

Table 2

13) The title says N=898. However, none of the variables have N=898. It might be important to explain what was considered as a missing variable and how were those variables treated? RESPONSE: We did not allow any missing values when calculating the total or subscales. Therefore, N is less than 898 in every scale. We remove the "(n=898)" from the title. In addition, we added this description to the statistical methods section. 

14) Line 320: Please correct the typo 

RESPONSE: We corrected the sentence. 

15) Line 322: In this study, self-stigma is the dependent variable, but the writing looks the opposite. 

RESPONSE: We corrected the sentence as follows;” If respondents had higher mean values with depressive symptoms, they experienced a higher level of self-stigma”. 

16) Line 325-327: Which data table does this statement refer to? 

RESPONSE: For this statement there was no separate table, so this result is described in the text only.

17) Line 350-352 The meaning of this sentence is not clear. 

RESPONSE: We clarified the sentence as follows;” Therefore, we did continue to examine this topic and found that the diagnosis of affective disorder was associated with higher scores of self-stigma than the psychotic disorder was.”

18) Limitation: Some of the limitations are against the main rationale of this manuscript and some are not relevant. I recommend removing the first and fourth limitation. Fifth limitation needs more elaboration explaining how psychosocial factors could have impacted on experiences of self-stigma. Last limitation could be the recommendation from this study. 

RESPONSE: We modified the limitations section as follows; 

“Our study has several limitations. First, the data collection was based on respondents’ self-reported information, so, precise information about diagnoses, treatment history and other objective measures could not be collected. We may therefore question the validity of the respondents’ own categorization of their diagnostic group. We can also speculate whether participants even had the insight to identify themselves in specific diagnosis category. For example, 17 patients responded with “Other” if they felt that they did not belong to either diagnostic group or if they were unsure to which diagnostic group they belonged. Indeed, it has been shown that some persons with mental illness may not be aware of their own diagnosis or that they lack insight on their illness. We did not use a specific assessment method to assure respondents’ mental status or insight for screening purposes during patient recruitment. It would have been helpful to have been able to confirm patients’ capacity or insight; not having been able to caused a potential point of risk. On the other hand, the service system in the study organization is divided into different pathways based on diagnostic groups: one for those with affective disorders and one for those with psychotic disorders. Respondents’ categorization of their diagnoses could then be assured by comparing organization records. 

Second, as the study was cross-sectional, results could only be captured at specific time points. To detect any causal or long-term effects of self-stigma in persons with mental disorders, further studies are needed to answer these questions. Third, this study focused participants’ socio-demographic and clinical factors, and lacked an investigation into the association between self-stigma and any psychosocial factors as been recommended by other investigators [11, 18]. To offer a deeper insight into this complex phenomenon, factors related to self-stigma, different career pathways, life stages and psychological factors, such as hope, self-esteem, empowerment and quality of life, should be studied in the future. 

Taken all together, despite the limitations, the findings from this study expand current knowledge about self-stigma and its related factors among patients living and treated in a community setting. A more context-oriented research approach with a longitudinal design could bring new understanding to how stigmatized attitudes develop as part of social integration and individual functioning and in specific contexts. Future work should also focus on further exploring the occurrence of self-stigma among vulnerable risk groups that are often excluded in society.”

19) Conclusion: line 424-425 in irrelevant in conclusion

RESPONSE: Sentence was removed.

---

## [Decision Letter · Decision Letter 1]

6 Apr 2022

PONE-D-21-27914R1Self-stigma among clients of outpatient psychiatric clinics: A cross-sectional surveyPLOS ONE

Dear Dr. Välimäki,

Thank you for submitting your manuscript to PLOS ONE. After careful consideration, we feel that it has merit but does not fully meet PLOS ONE’s publication criteria as it currently stands. While reviewer #2 is completely satisfied with your revision, reviewer #3 raises some additional questions. Therefore, we invite you to submit a revised version of the manuscript that addresses the points raised during the review process.

We look forward to receiving your revised manuscript.

Kind regards,

Stephan Doering, M.D.

Academic Editor

PLOS ONE

Journal Requirements:

Reviewers' comments:

Reviewer's Responses to Questions

**Comments to the Author**

1. If the authors have adequately addressed your comments raised in a previous round of review and you feel that this manuscript is now acceptable for publication, you may indicate that here to bypass the “Comments to the Author” section, enter your conflict of interest statement in the “Confidential to Editor” section, and submit your "Accept" recommendation.

Reviewer #2: All comments have been addressed

Reviewer #3: All comments have been addressed

2. Is the manuscript technically sound, and do the data support the conclusions?

Reviewer #2: Yes

Reviewer #3: Yes

3. Has the statistical analysis been performed appropriately and rigorously? 

Reviewer #2: Yes

Reviewer #3: No

4. Have the authors made all data underlying the findings in their manuscript fully available?

Reviewer #2: Yes

Reviewer #3: No

5. Is the manuscript presented in an intelligible fashion and written in standard English?

Reviewer #2: Yes

Reviewer #3: Yes

6. Review Comments to the Author

Reviewer #2: The authors have addressed the queries raised for the betterment of the manuscript. All the comments have been addressed well. One suggestion regarding type of mental disorder (affective, psychotic and others), please mention in patient recruitment section that few (2%) of the respondents had other than the affective and psychotic disorders.

Reviewer #3: Comments to author(s)

It is meaningful to study Finnish self-stigma with a large sample size. Through the revision of the manuscript, logical and persuasive power were strengthened. I would like to make a strong suggestion in the analysis section to increase the power of results. Analysis of differences in self-stigma scores according to the characteristics of current participants provides only little information to the reader. Regression models should be built and verified through correlation analysis and multivariate regression analysis to analyze factors affecting self-stigma using such a large number of samples.

Pleased, designate the dependent variable as self-stigma and show the model fit of explanatory power when the personal factors and clinical factors (eg. Depression) are used as independent variables. The author(s) need to perform advanced statistics with the help of a statistician to further solidify their results.

7. PLOS authors have the option to publish the peer review history of their article (what does this mean?). If published, this will include your full peer review and any attached files.

Reviewer #2: **Yes: **Bimala Panthee

Reviewer #3: No

---

## [Author Response · Author response to Decision Letter 1]

27 Apr 2022

REVIEWERS’ COMMENTS AND RESPONSES: 

Reviewer #2: The authors have addressed the queries raised for the betterment of the manuscript. All the comments have been addressed well. One suggestion regarding type of mental disorder (affective, psychotic and others), please mention in patient recruitment section that few (2%) of the respondents had other than the affective and psychotic disorders.

RESPONSE: Thank you for your valuable comment, which is related to the generalizability of the results. We added the sentence to the Discussion section as follows.” In addition, we had only 2% of patients who were diagnosed with other diagnoses than affective disorder or psychotic disorder. Therefore, our study results might be generalized in these two specific study groups only. “ 

Reviewer #3: Comments to author(s)

It is meaningful to study Finnish self-stigma with a large sample size. Through the revision of the manuscript, logical and persuasive power were strengthened. I would like to make a strong suggestion in the analysis section to increase the power of results. Analysis of differences in self-stigma scores according to the characteristics of current participants provides only little information to the reader. Regression models should be built and verified through correlation analysis and multivariate regression analysis to analyze factors affecting self-stigma using such a large number of samples.

Pleased, designate the dependent variable as self-stigma and show the model fit of explanatory power when the personal factors and clinical factors (eg. Depression) are used as independent variables. The author(s) need to perform advanced statistics with the help of a statistician to further solidify their results.

RESPONSE: Thank you for the comment for multivariable modelling. Actually we did perform multivariable modelling for total self-stigma, but unfortunately we forget to document that in statistical methods section. However, we conducted now multivariable modelling for total and subscales. Naturally there was some associations between the explanatory variables and therefore some explanatory variables were dropped from the final models (we first include all significant factors from univariate approach and then removed non-significant one by one). We added table 4 to illustrate multivariable models, and table 5 to show strong association between PHQ-9 and mental disorder categories. 

In the text we added the multi-way analysis of variance (ANOVA) in abstract and corrected the statistical section as follows: “Modelling was continued with multi-way ANOVA including all significant factors from univariate approach. Then non-significant terms were removed from the model one by one “. We clarified the results section for multivariable approach as follows: “Further, when the final model in multivariable approach was conducted, we found that length of mental disorder (p = 0.018) and PHQ-9 (categorized) (p < 0.001) were significantly associated to self-stigma. This means that the longer the respondent had had the mental disorder and the more severe their depressive symptoms were, the higher the total score of self-stigma was (Table 4). Similarly, multivariable models were programmed to self-stigma subscales, and the results were quite in line with univariate approach. The greatest exception was that type of mental disorder dropped from two final models due strong association with PHQ-9 (Table 5).” 

We have also taken this into account in the section of Discussion with sentence as follows: “While severity of the depressive symptoms and PHQ-9 were strongly associated so that in affective disorder the severity was more severe, PHQ-9 seemed to be stronger factor when total self-stigma was modelled.” 

In addition, reference (25) has been updated to year 2021.

---

## [Editor Report · Decision Letter 2]

23 May 2022

Self-stigma among clients of outpatient psychiatric clinics: A cross-sectional survey

PONE-D-21-27914R2

Dear Dr. Välimäki,

We’re pleased to inform you that your manuscript has been judged scientifically suitable for publication and will be formally accepted for publication once it meets all outstanding technical requirements.

Kind regards,

Stephan Doering, M.D.

Academic Editor

PLOS ONE

---

## [Editor Report · Acceptance letter]

3 Jun 2022

PONE-D-21-27914R2 

Self-stigma among clients of outpatient psychiatric clinics: A cross-sectional survey 

Dear Dr. Välimäki:

I'm pleased to inform you that your manuscript has been deemed suitable for publication in PLOS ONE. Congratulations! Your manuscript is now with our production department. 

Kind regards, 

on behalf of

Professor Stephan Doering 

Academic Editor

PLOS ONE